# Evaluating the Impact of Needle-Free Delivery of Inactivated Polio Vaccine on Nigeria’s Routine Immunization Program: An Implementation Hybrid Trial

**DOI:** 10.3390/vaccines13050533

**Published:** 2025-05-16

**Authors:** Diwakar Mohan, Mercy Mvundura, Sidney Sampson, Victor Abiola Adepoju, Garba Bello Bakunawa, Chidinma Umebido, Adachi Ekeh, Joe Little, Catherine Daly, Christopher Morgan, Sunday Atobatele, Paul LaBarre, Elizabeth Oliveras

**Affiliations:** 1Department of International Health, Johns Hopkins Bloomberg School of Public Health, Baltimore, MD 21205, USA; 2PATH, Seattle, WA 98103, USA; mmvundura@path.org (M.M.); jlittle@path.org (J.L.); 3Sydani Group, Abuja 900001, Nigeria; sidney.sampson@sydani.org (S.S.); aekeh@nigeriahealthwatch.com (A.E.); sunday.atobatele@sydani.org (S.A.); 4Jhpiego, Abuja 900271, Nigeria; victor.adepoju@jhpiego.org (V.A.A.); chidinma.umebido@jhpiego.org (C.U.); 5National Primary Health Care Development Agency, Abuja 900288, Nigeria; garba.bakunawa@nphcda.gov.ng; 6PharmaJet, Golden, CO 80401, USA; catherine.daly@pharmajet.com (C.D.); paul.labarre@pharmajet.com (P.L.); 7Jhpiego, Baltimore, MD 21231, USA; christopher.morgan@jhpiego.org (C.M.); elizabeth.oliveras@jhpiego.org (E.O.)

**Keywords:** acceptability, cost, dose sparing, effectiveness trial, feasibility, fIPV, fractional dose, implementation research, intradermal, IPV

## Abstract

**Background/Objectives**: The Tropis^®^ ID device (PharmaJet^®^), a needle-free injection system, is a World Health Organization prequalified, hand-held device, which delivers intradermal injections without the use of needles and has previously been used for the delivery of fractional doses of inactivated polio vaccine (fIPV) in campaign and house-to-house settings. This implementation research study aimed to comparatively evaluate the vaccine coverage, cost, feasibility, and acceptability of using Tropis for fIPV for routine immunizations in two states in Nigeria (Kano and Oyo). **Methods**: The study included: (i) a cluster randomized trial (22 intervention facilities using Tropis for fIPV and 30 control facilities using the standard of care [SoC—full-dose IPV]) to assess the effectiveness in terms of improving the coverage of two doses of IPV, using a coverage survey involving 3433 children (aged 3–12 months); (ii) a pre- and post-implementation micro-costing evaluation involving the intervention facilities to estimate the costs; and (iii) mixed methods assessments (post-training assessment, provider survey, key informant interviews, and focus group discussions) to assess the feasibility and acceptability of fIPV delivery using Tropis. **Results**: The intention-to-treat analysis among the 3433 children surveyed did not show any difference between the intervention and control groups, primarily due to low compliance (approximately 50% of target beneficiaries reported Tropis use). The more relevant per protocol analysis, adjusting for lower compliance, showed that among those vaccinated with Tropis, second dose IPV coverage was 11.2% higher than the SoC. The delivery of fIPV using Tropis compared to the SoC resulted in incremental program cost savings, ranging from USD 0.07 to USD 1.00 per dose, administered across the scenarios evaluated. High acceptability was seen amongst caregivers (94%), and 95% of healthcare workers preferred Tropis over the SoC. **Conclusions**: Tropis is effective, feasible, acceptable, and saves costs when used as part of routine immunization programs.

## 1. Introduction {6a: Background and Rationale}

Since the Global Polio Eradication Initiative was launched in 1988, the global occurrence of polio has decreased by 99.9% [1]. Today, wild poliovirus (WPV) is endemic in just two countries, Afghanistan and Pakistan [2]. Nigeria was declared free of WPV in August 2020 [3] but is classified by International Health Regulations as a state infected with circulating vaccine-derived poliovirus type 2 (cVDPV2) [4]. The inactivated polio vaccine (IPV) is an important immunization tool for preventing paralysis as a result of both WPV and cVDPV, and has been shown to be effective in boosting mucosal protection in individuals previously immunized with the oral polio vaccine (OPV) [5]. Beginning in 2013, the World Health Organization (WHO) recommended that at least one dose of IPV be introduced into all routine immunization (RI) schedules [6]. Since then, based on findings that a second dose of IPV confers higher immunogenicity, the Strategic Advisory Group of Experts (SAGE) has recommended the introduction of a second dose of IPV as a next step toward the eventual complete withdrawal of bivalent OPV [7].

IPV can be delivered as a full dose into the intramuscular (IM) region or, alternatively, as a fractional (one fifth [20%]) dose delivered into the intradermal (ID) compartment. This lower dose administered intradermally has been found to be safe and immunogenic, with no substantial difference in seroconversion between two and three doses of fIPV, and two and three doses of full-dose IPV [8]. To date, six countries (Bangladesh, Cuba, Ecuador, India, Nepal, and Sri Lanka) have adopted fIPV for RI [9]. The use of fIPV can reduce the cost of IPV procurement for an immunization program and allow more children to be vaccinated using a given supply [10].

Nigeria’s RI schedule includes two full doses of IM IPV, administered to infants at 6 and 14 weeks of age. While Nigeria improved its full antigen coverage from 33% in 2016 to 57% in 2022 [11], the immunization program expressed interest in assessing innovations that could further improve coverage and reduce costs, as the country prepares for the transition from Gavi support [12]. Nigeria is currently using fIPV during the administration of IPV for Polio Supplemental Immunization Activities/campaigns, as recommended by the Nigeria Immunization Technical Advisory Group. However, there have been concerns by the health authorities about the fidelity of the dose and depth into the ID region when fIPV was delivered with a needle and syringe, because of the training required to correctly provide the ID injection [13].

The Tropis^®^ ID device (PharmaJet^®^, Golden, CO, USA), a needle-free injection system, which has secured WHO prequalification, is an ID delivery method that has been successfully used in campaign and house-to-house settings in high-risk polio environments [14]. Tropis has been found to be highly acceptable for fIPV delivery [14,15,16,17] and can increase immunization coverage [15,16]. We hypothesized that the use of Tropis for fIPV delivery as part of RI could increase the uptake of the second dose of IPV if it is well-received by families and acceptable to healthcare workers. We also sought to generate evidence on the feasibility and cost of its introduction, as these data would be informative for decision making for the scaling up of the technology.

### Objectives {7: Objectives}

Our theory of change included the hypothesis that increased acceptance and value attributed to a needle-free option would encourage greater uptake of the second dose of IPV and, subsequently, that if this option proved feasible in routine settings, the delivery of fractional doses would enable significant cost savings.

In line with the above logic, this study had three main objectives:To evaluate the effectiveness of Tropis for fIPV delivery as compared to the standard of care (SoC, 0.5 mL IM injection with a traditional needle/syringe) in regard to improving IPV2 coverage among children aged less than one year;To assess the incremental cost in regard to the immunization program of using Tropis for fIPV delivery as compared to standard vaccination practice;To understand the feasibility and acceptability of fIPV delivery using Tropis.

## 2. Materials and Methods

### 2.1. Study Design {8: Trial Design}

We designed an implementation research study, which included the following elements:A cluster randomized trial involving a coverage survey of the target population (children 3–12 months old) to assess the coverage of IPV using Tropis compared to the SoC;A pre- and post-implementation micro-costing study, with primary and secondary data collection from the health facilities implementing fIPV using Tropis, to estimate the immunization program costs for fIPV versus the SoC;Mixed methods assessments (post-training assessment, provider survey, key informant interviews (KIIs), and focus group discussions (FGDs)) to assess the feasibility and acceptability of fIPV delivery using Tropis.

### 2.2. Study Settings {9: Study Setting}

Nigeria is a federation with 36 states and the Federal Capital Territory (Abuja) is divided into 774 Local Government Areas (LGAs) [18]. The country has a pluralistic health system, comprising of public and private providers and traditional and modern care systems [19]. All three tiers of the government (federal, state, and local) collectively share the responsibilities for providing health services and programs [19]. Services are provided through primary, secondary, and tertiary entities [20]. Vaccination services are predominately offered at health facilities.

This study was implemented in two states, Kano in the north and Oyo in the south of the country, to balance the local stakeholder expectations in Nigeria. Kano has 44 LGAs and Oyo has 33 [21,22]. Both states were selected to receive the intervention based on the following pre-determined criteria: vaccination coverage for the third dose of the pentavalent vaccine from 40% to 60% (as a measure of the strength of the vaccination program), geographically accessible, conflict-free operating environment, and absence of other fIPV programs during the intervention period.

In consultation with key stakeholders, three LGAs (Gwale, Kabo, Kura) in Kano and four in Oyo (Atiba, Ibadan North, Iseyin, Ogo Oluwa) were purposively selected. This selection was based on the urban–rural composition and safety of the operating environment. *{10: Eligibility criteria for clusters}.* Within each LGA, the smallest administrative unit is a ward. Each ward contains one apex health facility. This apex facility, with its catchment area, was the unit of randomization (cluster).

### 2.3. Intervention Description: Tropis {11a: Intervention Description}

PharmaJet’s Tropis is a WHO-prequalified, hand-held, needle-free ID delivery device that can be used for fIPV administration [23]. It delivers a spring-powered injection in one-tenth of a second, through a narrow stream of fluid that penetrates the skin, with a precise dose and depth (Figure 1) [24]. A 0.1 mL fixed dose is delivered without generating sharps waste [24]. No external power source is required for its function [24]. Tropis can improve the patient and caregiver experience, resulting in increased vaccine compliance [24], and enables ID dose sparing and associated cost savings.

### 2.4. Randomization and Intervention Allocation

Of the 52 apex facilities selected with their catchment areas, we conducted the randomization using an unequal allocation ratio, with 22 intervention and 30 control facilities, to optimize the statistical power within the project constraints. The randomization was stratified based on the state, urban/rural classification of the LGAs and the size of the facilities, based on the previous annual volume of immunizations. Because this was a cluster randomized trial, the randomization and allocation were conducted at one time at the start of the study *{16a—Sequence generation and allocation of intervention}*. Study arm assignments were unblinded due to the nature of the intervention *{17b: blinding}*.

In the intervention facilities (a total of 22, with 11 per state), children under one year of age received fIPV intradermally, during routine immunization service provision. Tropis was used to administer fIPV through the ID route at 6 and 14 weeks. ID delivery is not the standard of care for routine immunization programs but has been approved by the Nigerian National Primary Health Care Development Agency (NPHCDA) for use in IPV campaigns. In the control wards (a total of 30, with 15 per state), immunization with IPV was provided in line with the existing standard of care (0.5 mL of IM injection using a traditional needle and syringe) *{6b: Explanation for choice of comparator}*.

### 2.5. Description of the Implementation

Three staff from each intervention facility participated in a two-day training that included the basics of IPV and practical training on the use of the Tropis device. All the children residing in the defined catchment areas or those accessing the intervention apex facility for immunization services, due for IPV1 during approximately the six months of the implementation, were eligible to receive the intervention *{10: Eligibility criteria}*. The delivery of fIPV using Tropis was co-administered with other vaccines during routinely scheduled vaccination sessions held at the health facilities or in the community (outreach). Caregivers were educated about Tropis at the time of vaccination of their infants. Community-based volunteers performed their regular duties to generate demand for health services in the intervention and control areas. No additional demand generation activities or messaging were introduced for the delivery of fIPV using Tropis in regard to either the intervention or control sites during this study. Caregivers who refused to have their child vaccinated using Tropis were provided with SoC vaccination. Healthcare providers logged the details of the participants that opted out from receiving the intervention. Due to the nature of the intervention being tested, blinding of the participants (providers and caregivers of the children) was not possible.

### 2.6. Survey to Measure Immunization Coverage

We used a household survey, conducted more than a month after the 6 months of the implementation had ended, of the children aged 3 to 12 months to measure the coverage in the intervention and control wards.

From the catchment area of each apex facility, we selected a sample of children based on a two-stage design process. Within each ward, we contacted health system personnel and local government workers to create lists of communities (villages in rural and wards in urban LGAs) and their resident populations that receive services (fixed or outreach) from the apex facility. These lists were used to create enumeration areas of approximately 1500 people. This was based on the assumption that children aged 3 to 12 months composed approximately 2.5% to 3% of the overall population in Nigeria. Large communities were split along identifiable boundaries, while smaller adjacent communities were combined. Finally, from each ward, five enumeration areas were selected using the probability proportional to size (PPS) methodology [25]. Within each enumeration area, all the households were visited and screened for eligible children. If the population of the catchment area of the apex facility was less than 7500 people, all the households were enumerated for the survey.

#### 2.6.1. Survey Procedures {8a: Plans for Assessment and Collection of Outcomes. 19: Data Management}

The household surveys comprised two sequential procedures. First, the household head provided information on the relevant socioeconomic characteristics, composition of the household, presence of children aged between 3 to 12 months, and identified the mother or caregiver of the child(ren). Second, if eligible child(ren) were identified in the household, then the mother or caregiver was asked to provide information on their vaccination status, child’s immunization history, and experience with immunization services. If more than one child was eligible, then this process was repeated for each child living in the household.

The interview teams were trained to obtain consent in the local language (Hausa in Kano and Yoruba in Oyo). After presenting themselves to the household head, the interviewer explained the study purpose and procedures and requested oral consent ***{26a: Informed Consent}***. If oral consent was given, the interviewer then explained the study to the mother or caregiver and requested written consent. If one or more age-eligible child resided in the household, the interviewer asked permission from the mother or caregiver to obtain each child’s vaccination information through the use of their vaccination card. If the card was not available, a series of questions was asked to assess their vaccination history. If consent was not obtained, the interviewer thanked the household for their time and moved on to the next household. If the household head or eligible caregiver was not available, three attempts to revisit the household were made before reporting a non-response.

#### 2.6.2. Sample Size and Power {14: Sample Size}

Our sample size assessments were predicated on detecting a 10% increase in IPV2 coverage (absolute difference) in the intervention clusters (catchment area of the apex facility) compared to the control. We assumed the most statistically conservative scenario of achieving 50% vaccination coverage. Overall, we targeted a sample of 5200 mothers or caregivers (intervention: 2200; control: 3000) to be interviewed for the survey, assuming a refusal rate of 5%, a power of 80%, a type I error of 0.05, and an intraclass correlation coefficient (ICC) of 0.1 at the cluster level. At the end of the survey, we achieved 66% of our planned overall sample. Our original assumptions were based on a very conservative 50% base prevalence, and high ICC and refusal rates. Based on our final control site prevalence of 63.4%, low refusal rates and an ICC (with stratification) of 0.04, the final sample of 3433 had a power of 80% to detect a difference of at least 8.6% in the outcome.

#### 2.6.3. Outcomes {12: Outcomes}

The primary outcome was the percentage of children aged 3 to 12 months who received two doses of IPV. The secondary outcome was the percentage of children aged 3 to 12 months who received at least one dose of IPV. The receipt of an IPV dose was determined by the vaccination card or reported by the caregiver. We assessed the compliance in regard to the treatment via a survey question that asked caregivers if they were present during the immunization session and observed the vaccine administration using Tropis. A photo was shown to caregivers for verification. Other outcomes of interest (intervention area only) included mothers or caregivers reporting the acceptability of Tropis, which was assessed as an implementation outcome for Aim 3.

#### 2.6.4. Data Analysis of Coverage Survey {20a: Statistical Methods for Primary and Secondary Outcomes}

The effectiveness estimate for the intervention is defined as the change in coverage for the second dose of IPV among the children of the respondents. We estimated the absolute and relative difference in coverage for the inactivated polio vaccine according to the dose number (IPV1 and IPV2). All the estimates were adjusted for the survey design, accounting for the stratification (urban/rural stratification of LGA, facility size) for the variables used in the randomization. Generalized linear models, with the above indicators, were used to assess the estimates adjusted for clustering at the community level, using clustered standard errors. We conducted sub-group analysis to look for heterogenous effects according to the household’s baseline characteristics, such as education, wealth, and place of residence.

Our primary analysis was based on a modified intention-to-treat approach, with the treatment variable based on randomized allocation at the apex facility level. Given our proposed theory of change, the per protocol analysis of the infants who received Tropis vaccination (who complied with the intervention) was identified as being as important to provide a compliance-adjusted effect. We performed a per-protocol (PP) and an as-treated (AT) analysis given the reduced levels of compliance in regard to the intervention [26]. The PP analysis analyzed data from those that complied only (participants who followed the randomized allocation, and excluding those who were non-compliant), respondents who reported Tropis use in the intervention arm and the SoC in the control arm were part of the analysis. The AT analysis considered the treatment received by the participant, whereby the respondents who reported Tropis use and no use of Tropis, irrespective of the random assignment, were part of the analysis. Both of these analyses were restricted to caregivers who reported being present at the time of immunization. To account for any imbalance between the groups in regard to the baseline characteristics, both the PP and AT estimates are presented with 95% confidence intervals (CIs), with adjustment for the baseline characteristics of the respondents, including education, wealth category, polygamy, place of birth, and distance from health facility.

### 2.7. Costing Analysis

We conducted a pre- and post-implementation micro-costing study, with primary data collection and records extraction from the 22 intervention health facilities in Kano and Oyo, augmented with secondary data and assumptions to inform the scenario modeling. The study included a time-and-motion analysis that compared the time taken by vaccinators to administer IPV as part of the SoC during the pre-implementation process for fIPV and repeated again during the implementation of fIPV using Tropis. The data collectors observed and recorded the time spent by vaccinators with each child, starting the timing when the vaccinator prepared the IPV dose, until disposal of the needle and syringe sharps waste in the safety box (SoC) and the disposal of the needle-free syringe into a container (no sharps waste). Observations were conducted during one session at each intervention facility in regard to the provision of the SoC, in July 2023, and then, repeated again, during the implementation of fIPV using Tropis, in November 2023.

We used structured costing questionnaires to collect primary data from the intervention health facilities. Data collectors interviewed immunization staff at the intervention health facilities to quantify the resources used for immunization program activities, including vaccine procurement, service delivery, vaccine distribution and storage, and other routine program activities.

The vaccine procurement and vaccine delivery costs evaluated included: (1) vaccine and supplies procurement costs, including costs of IPV vaccines and injection devices (traditional needle and syringe for the SoC, and Tropis devices, syringes, and vial adapters for the delivery of fIPV using Tropis); (2) vaccine storage and distribution costs, including costs of cold chain storage for vaccines and transport costs for distributing vaccines between the different levels of the supply chain; (3) service delivery costs, including vaccinator time costs and transport costs of the vaccination team to get to the outreach locations; (4) costs of other routine program activities, including human resource time costs for activities such as program planning and social mobilization, and the costs of resources used for waste management of the immunization materials; and (5) introduction costs for needle-free fIPV, mainly training of healthcare workers to use Tropis.

The data for the pre-implementation period were collected in July 2023 and, for the post-implementation, in November 2023. The costing was conducted from the immunization program perspective. The time horizon for the costing was the 6-month period, aligned with the implementation phase. We used Stata (version 18) and Microsoft Excel (Microsoft 365 Apps for enterprise, Version 2503) for the analysis. Salary scales for staff working on the immunization program, which were used to value human resource time, were collected from government documents, and replacement prices for equipment and vehicles were obtained from online databases. The annualized costs of capital were calculated assuming a 10-year economic life for vehicles and cold chain equipment, with the economic life of such goods discounted at a rate of 3%. Since IPV vaccine activities were integrated with other routine vaccines, the costs were allocated based on the proportional quantities of IPV vaccines relative to other vaccines. For the Tropis devices, which are reusable, we assumed a useful life of 5 years. The unit costs for each of the resources used were multiplied by the quantities used to estimate the costs. The main metric estimated was the cost per dose administered. We calculated this as the volume weighted mean cost per dose administered:$$Volume\;weighted\;mean\;cost\;per\;dose\;=\;\frac{\sum_{i=1}^{n}{Cost}_{i}}{\sum_{i=1}^{n}{Doses}_{i}}$$
where i is each health facility in the study sample and n is the sample size. The costs are the total costs for each health facility and the doses are the number of IPV doses administered (number of full doses for the SoC and number of fractional doses for the intervention). The costs are reported in 2023 US dollars. An exchange rate of 426 Naira per US dollar was used.

Where data gaps existed or where uncertainty existed, we conducted scenario analysis to test the robustness of the estimates generated from the costing study. Specifically, we used assumptions to inform the wastage rates used in the analysis due to poor quality program data, with the values used shown in Table 1. We evaluated various scenarios to understand the impact of the inputs on the delivery cost estimates. The immunization program data on the number of doses administered under the SoC showed lower doses administered than fIPV and, so, we evaluated a SoC scenario wherein we held the doses administered constant between the SoC and fIPV (shown as the SoC full-dose scenario in the results graphs), which would make the SoC more favorable given the inverse relationship between the number of doses and cost per dose administered.

In regard to the study settings, the prices of Tropis devices, syringes, and adapters were USD 415, USD 0.59, and USD 0.45, respectively, and the length of the training for the vaccinators to use Tropis was 2 days. We explored fIPV scenarios by varying the number of Tropis devices per health facility, by shortening the vaccinator training duration, and by varying the procurement prices for Tropis devices and syringes, as higher volume procurement results in unit cost reductions for Tropis devices and supplies. In the first scenario (named fIPV scenario 1 in the results), we assumed that each facility would receive two Tropis devices, as in the study settings, but with lower prices for the devices, syringes, and adapters (USD 374, USD 0.50, and USD 0.41, respectively), and held the length of training of the vaccinators to use Tropis at 2 days, as conducted in regard to the study settings. In the second scenario (named fIPV scenario 2 in the results), we assumed that 75% of the facilities would receive one device and 25% would receive two devices (higher throughput facilities) and that the prices of the devices, syringes, and adapters were USD 291, USD 0.34, and USD 0.38, respectively, and a reduction in the length of training of the vaccinators to use Tropis devices to 1 day. In the third scenario (named fIPV scenario 3 in the results), we assumed that one device was available for all the facilities, as well as the same prices as in scenario 2, and also a reduction in the length of training of the vaccinators to use Tropis devices to half a day.

Table 1 shows some of the key unit prices and assumptions for the costing analysis.

We also conducted a modeling analysis to evaluate the cost savings to the immunization program if they were to switch to fIPV delivery using Tropis in the whole country. Using data from 2023, which showed that approximately 13 million full IPV doses were procured in Nigeria, we estimated the savings if fIPV were used, assuming 75% of the health facilities received one Tropis device and 25% received two devices, with the latter being high-volume facilities. We used the large volume procurement costs for Tropis devices in this analysis (USD 291 for devices, USD 0.34 for syringes, and USD 0.38 for adapters).

### 2.8. Feasibility and Acceptability Assessment

We used mixed methods to assess the feasibility of using Tropis for RI and its acceptability to both caregivers and healthcare workers at all levels (community, facility, LGA, state, and federal). All healthcare workers trained in the use of Tropis were invited to participate in a post-training assessment prior to the start of the implementation and a survey during the fifth month of the implementation. We also conducted KIIs with a sample of healthcare workers; first, we selected the facilities for the qualitative component, based on accessibility and facility size (a mix of high- and low-volume facilities); then, we selected the healthcare workers based on their role, years of experience in regard to the RI program, and their knowledge of the study. Respondents at state and LGA levels were eligible if they had attended the Tropis healthcare worker training. To understand the caregivers’ views, we included questions in the household survey for this purpose and conducted FGDs with the caregivers of children who received at least one dose of IPV using a Tropis device.

Quantitative data were analyzed using descriptive statistics and qualitative data were analyzed through the use of an iterative thematic analysis process. Themes were identified from our pre-existing hypothesis and transcripts were also scanned for emerging themes; these themes were assigned codes for the analysis of dominant and unique themes in all the transcripts. Double coding was conducted on five transcripts per state to ensure that the coders were consistent in their application of the codes. The coded data were reviewed by the study team and categorized under major themes that addressed the study questions.

## 3. Results

The household survey was completed in Kano and Oyo, from February to June 2024, following almost six months of implementation. A total of 97,165 households were visited by the data collectors and screened for children in the age range from 14 weeks to 12 months. We assessed the vaccination history of 3433 children, 1903 in Kano and 1530 in Oyo (Figure 2). The baseline characteristics of those included in the intervention and control arms are provided in Appendix A.

Compliance in regard to the intervention was based on the receipt of vaccinations using Tropis, as reported by caregivers who were present during the immunization service visit. About 89% of the caregivers reported being present when their children received their vaccination. Gradients were observed based on wealth and education, but were not statistically significant (Appendix A). The caregivers present at the vaccination were asked if their child had been vaccinated using Tropis. The proportion of the population receiving IPV using Tropis in the intervention arm was 50% (compliance) and 5.6% in the control arm (contamination). The proportion was higher in Oyo (62.2%) compared to Kano (39.6%). There was an increase in the proportion of children receiving IPV via Tropis from the poorest to the richest wealth quintiles and the proportion was higher among monogamous and urban households.

### 3.1. Intention-to-Treat (ITT) Estimate

Among those living in the catchment areas of the health facilities randomized to provide IPV delivery using Tropis (intervention) or the SoC (control), the prevalence of IPV2 was 63.8% (58.9–68.8%) in the SoC arm and 61.7% (56.4–67.0%) in the Tropis arm. The estimates were 68.0% (61.5–74.4%) and 63.2% (56.9–69.4%) for Kano, and 58.7% (52.4–65.0%) and 59.9% (50.9–68.9%) for Oyo, respectively (Table 2). The ITT estimates restricted to children older than 18 weeks of age had slightly higher estimates for IPV2 prevalence, but these were not significantly different.

### 3.2. Per-Protocol Estimate

Among those who complied with the treatment (reported receiving IPV delivery using Tropis in the intervention arm and receiving the SoC in the control arm), the prevalence of IPV2 was 72.5% (67.6–77.3%) in the SoC arm and 82.3% (76.4–88.2%) in the Tropis arm. These estimates were 76.7% (71.5–82.0%) and 89.2% (82.7–95.7%) for Kano, and 67.0% (60.1–74.0%) and 77.2% (69.9–84.4%) for Oyo, respectively (Table 3). The per-protocol estimates restricted to children older than 18 weeks of age had slightly higher estimates for IPV2 prevalence and showed a similar increase in IPV2 prevalence in the intervention arm over the control arm.

The regression models produced treatment estimates adjusting for the LGA, education, and gender of the head of the household, polygamous household, wealth, urban/rural residence, facility size, and place of delivery (Table 4). The ITT model shows a difference in the prevalence of IPV2 of −4.3% (−10.3–1.7%), but the difference is not statistically significant. A similar model using the per-protocol analytical model shows a 10.9% (6.2–15.8%) increase in IPV2 administration in the intervention arm over the control arm. On a relative basis, the odds of IPV2 administration are not statistically different between the intervention and the control arms when the ITT sample is applied, while there is a doubling of the odds (43–178%) when restricted to the per-protocol sample. The facility size and distance from the household to the apex facility were not significantly associated with the receipt of IPV2. The as-treated estimates provide similar estimates as the per-protocol analysis. The coverage of other vaccines was assessed for the same populations. The coverage of Rotavirus doses did not differ significantly between the study arms, while the coverage of the third dose of the pentavalent (Penta3) and Pneumococcal vaccine mirrored the differences seen in regard to IPV2 (Appendix A).

### 3.3. Costing Analysis

For the SoC, a total of 214 full-dose IPV vaccinations at 21 intervention health facilities within the two study states were observed and the time taken was recorded. At the sessions observed, an average of 26 children were vaccinated per session. The time to give an IPV injection using a needle and syringe averaged 44 s and had a median of 35 s. For the intervention involving fIPV delivery using Tropis, a total of 138 fIPV vaccinations at 22 intervention health facilities within the two study states were observed. The vaccination sessions averaged 24 children per session and the time to give an fIPV injection using the Tropis device averaged 39 s and had a median of 33 s. There was no change in the practice of healthcare workers regarding how the sessions were organized.

We estimated that the cost per dose administered according to the SoC is higher than the cost per dose administered in terms of the needle-free fIPV, as per the study settings. This finding of a lower cost per dose administered in regard to fIPV versus the SoC occurred despite the relatively higher prices of Tropis devices and syringes given the low procurement volumes under the study settings. The incremental cost savings from dose sparing due to needle-free fIPV exceeded the cost increase associated with the procurement of Tropis syringes and devices. As shown in Figure 3, under the study settings, the incremental savings achieved due to needle-free fIPV administration were USD 0.20 per dose compared to the standard of care (full-dose scenario), assuming a 30% incremental wastage rate as a result of fIPV administration.

As shown in Figure 4, across all the scenarios evaluated, the cost per dose administered for the needle-free fIPV was always lower than the cost per dose administered for the full-dose IPV. The incremental savings as a result of the needle-free fIPV delivery ranged from USD 0.16 to USD 0.93 per dose compared to the standard of care (full-dose scenario), assuming a 30% wastage rate as a result of the fIPV administration.

When we varied the fIPV wastage rates, the incremental savings per dose for the needle-free fIPV administration ranged from USD 0.23 to USD 1.00 when the incremental wastage rate as a result of fIPV administration was assumed to be 20% and ranged from USD0.07 to USD 0.84 when the incremental wastage rate was assumed to be 40%.

The analysis of the procurement cost implications of switching to fIPV compared to the SoC showed that the immunization program could save approximately USD 50 million over a 5-year period (Table 5). The bulk of the savings would come from savings in regard to vaccine procurement costs, with these savings outweighing the incremental costs associated with procuring Tropis devices and supplies.

### 3.4. Feasibility and Acceptability

Tropis was the preferred method of vaccination for RI, as reported by 97% of the 65 healthcare workers who participated in the provider survey, 5 months into implementation. The main reasons for this included a belief that it was easier to use than the SoC needle and syringe (95%), and the perception that children experienced less discomfort or crying with the use of Tropis (94%). A positive caregiver response was the third most-cited reason (83%). All 65 healthcare workers who responded to the provider survey also reported that they were confident or very confident in using the device during both fixed and outreach sessions. The healthcare workers who participated in the qualitative interviews also highlighted the simplicity of operating the Tropis device, emphasizing that it required minimal training and expertise. They reported that unlike traditional needle and syringe methods, the Tropis device was user friendly and did not necessitate complex procedures for loading or administration. Additionally, healthcare workers in both the surveys and interviews felt the device positively impacted RI, as there was no vaccine wastage, and it could easily be used during outreach and fixed immunization days.

Tropis was highly acceptable amongst healthcare workers and caregivers. Of the 710 caregivers in the household survey whose children received at least one dose of IPV using Tropis, more than 96% (n = 685) said they would recommend it to a friend, and more than 96% (n = 682) said they would be more likely to return for future vaccinations if they knew Tropis was being used. In addition, 67.6% (n = 480) responded that they had permitted the healthcare worker to administer all the scheduled vaccinations because there was a needle-free option. The results from the FGDs with caregivers were consistent with these findings. When asked about their preferred mode of IPV vaccination, the respondents in all the FGDs preferred the Tropis device. Caregivers whose children had been vaccinated with the Tropis device consistently expressed satisfaction and happiness with the device, emphasizing its effectiveness in reducing the pain, discomfort, and adverse reactions associated with traditional needle and syringe administration. Caregivers were particularly pleased with the absence of crying and swelling impacting their children after immunization. Furthermore, respondents expressed a desire for the Tropis device and strongly advocated that the device be used to administer other vaccines.

## 4. Discussion

This study is one of the first effectiveness trials for Tropis in RI settings, and its implementation in Nigeria adds to the evidence on its value in the administration of IPV, specifically for RI programs. The cluster randomized design took into account the need for efficiency in terms of the resources allocated and optimal learning as part of a two-state implementation process. The sample for the outcomes covered almost 100,000 households across 52 apex facility catchment areas, across seven LGAs in Nigeria. The costing study found that administration of fIPV using Tropis results in cost savings from the health system perspective. The study also found that the use of Tropis was highly acceptable to vaccinators and caregivers and that it is feasible to integrate the use of Tropis into RI, even when other co-administered vaccines are being given using a needle and syringe.

The study found that only half of those expected to be covered by the intervention (those living in intervention clusters) received fIPV delivered using Tropis. The primary causes for non-compliance are likely the fact that only the largest (apex) facility in a catchment area was provided with Tropis devices, and that immunization was available to families through a variety of platforms. When restricting the analysis to those in the intervention area who did receive IPV administration using Tropis, the coverage of the second IPV dose was 11.2% higher in the intervention arm compared to the control arm (PP estimate). On a relative basis, the odds of receiving two doses of IPV are doubled when Tropis is reported as being used for vaccination administration. While the ITT is a more conservative estimate, the compliance adjusted PP estimate is a more direct test of our theory of change: that increased acceptability would make families more likely to return for a second dose.

Using the study settings and across all the modeled scenarios evaluated, needle-free fIPV administration resulted in a cost saving compared to the SoC. That is, the cost per dose administered was lower with needle-free fIPV versus the SoC. The incremental savings as a result of needle-free fIPV administration ranged from USD 0.07 to USD 1.00 per dose administered, across the scenarios evaluated. Our findings on the cost savings as a result of fIPV administered using Tropis compared to the SoC involving full-dose IPV are consistent with the findings from a prior modeling study [10]. The cost savings we estimated occur even though we used conservative assumptions on the extent of the useful life of Tropis devices, as we assumed 5 years of use, and yet health facilities with small catchment sizes would not reach the 20,000 injection capacity of Tropis devices within 5 years and, so, could benefit from a longer useful life. Thus, the cost savings could be larger than we estimated based on these conservative assumptions. We also found that if Nigeria were to switch from the SoC to needle-free fIPV, the amount spent on IPV procurement (for vaccines and injection devices) could be reduced by approximately USD 50 million over a 5-year period.

We did not conduct a cost-effectiveness analysis (CEA) because a CEA is only needed when an intervention is both more expensive and more effective than the comparator and, so, the CEA would determine whether the incremental cost associated with the intervention is worth the incremental health impact. In our study, we found that fIPV administration using Tropis did actually result in cost savings and was also more effective, as it improved vaccine coverage compared to the SoC. Therefore, a CEA was not needed as fIPV using Tropis outperformed the SoC.

Nigeria is currently using 10-dose vials of IPV, which translates to 50 fractional doses. If switching to fIPV, using a smaller vial size could be considered to reduce vaccine wastage. In addition, as the costing study showed, reducing the length of the training session from what was conducted in this study to shorter but still effective training could result in additional cost savings. Shorter training or on the job training should be explored when considering intervention scaling.

Tropis was highly acceptable to both the healthcare workers who used it and to caregivers whose children received fIPV using Tropis. Tropis was found to be user friendly and easy for providers to use. This is consistent with prior studies on Tropis. Yousafzai et al. (2017) reported that vaccinators in Pakistan found that the device was easy to fill and also reported finding that it was easy to administer vaccines using the device [27]. In the context of RI, healthcare workers generally felt that the use of Tropis had positive effects, including limiting vaccine wastage and improving coverage. Healthcare workers reported that caregivers whose children were immunized using Tropis expressed trust, satisfaction, and appreciation for the device’s efficacy, noting that their children did not appear to experience discomfort or distress during the vaccination process. They reported that many mothers appreciated the absence of adverse reactions, such as pain at the site of injection, following immunization using the device.

The findings also show that the Tropis device was highly acceptable to caregivers. This was evident from both the household surveys and FGD sessions with caregivers, and from the interviews with healthcare workers and other stakeholders. The key reasons for this were the absence of pain and minimal adverse events following immunization. This echoes past studies that have reported on the speed of administration, comfort of the device, and ease of use, as the drivers for accepting and preferring the Tropis device for vaccine administration [16,17,28]. Consequently, healthcare workers highlighted observing an increased number of caregivers bringing their children for immunization after the introduction of the Tropis device. This reiterates the level of acceptability of the Tropis device by caregivers within the communities surrounding the health facilities.

Our study has several limitations. Our results are based on a short implementation period (6 months), which may have affected the outcomes. The short duration may reflect the introduction of a novel approach, and the initial weeks or months were spent on learning and adjustment of the vaccinators. It also could be possible that some of the positive attitudes could decrease during a longer implementation period; this also did not allow for capturing the long-term use of the devices. That said, certain aspects of the study implementation, such as additional work undertaken by providers to track the eligibility for Tropis and the training of only three healthcare providers per clinic initially, created bottlenecks at high-volume clinics that may not reflect use in RI conditions.

A program initiated by a development partner in Kano during the study period incentivized vaccination with cash transfers. This may have reduced the number of children served through public outreach services, but may also have increased vaccination coverage in the control area; one of the most common reasons caregivers mentioned for vaccinating their child was the offer of monetary incentives. This would have attenuated any effects. 

This study relied on routine program data to provide vaccine wastage rates, but found that the data quality was poor. Thus, the costing analysis had to rely on assumptions for wastage rates. However, we still found that the administration of fIPV using Tropis resulted in a cost saving, even when assuming incremental wastage rates of up to 40% over those for the SoC. Future evaluations should plan to collect study-specific wastage rates, so the analysis is informed by program data.

## 5. Conclusions

The Tropis needle-free device has been demonstrated to be effective, feasible, acceptable, and results in cost savings, in regard to its use in Nigeria’s RI program for the delivery of fractional doses of IPV, alongside other co-administered vaccines. This adds to the evidence on the usefulness of the device previously established during studies of campaign use. With recent reductions in funding for global health programs (including the Global Polio Eradication Initiative), there is a need for innovations that reduce costs without reducing coverage. These study results can inform decision making on the role of innovative devices within national plans to strengthen routine immunization, and global strategies, such as Gavi 6.0.

## Figures and Tables

**Figure 1 vaccines-13-00533-f001:**
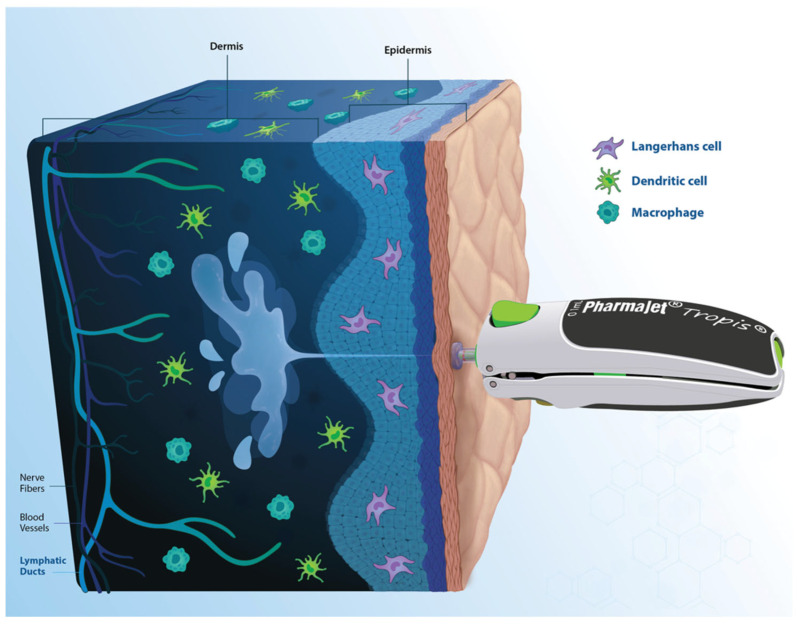
PharmaJet engineers have designed the Tropis^®^ ID needle-free system to target the intradermal compartment, which is rich in terms of the diversity and density of antigen-presenting cells, with a precise dose and penetration depth (device is not to scale).

**Figure 2 vaccines-13-00533-f002:**
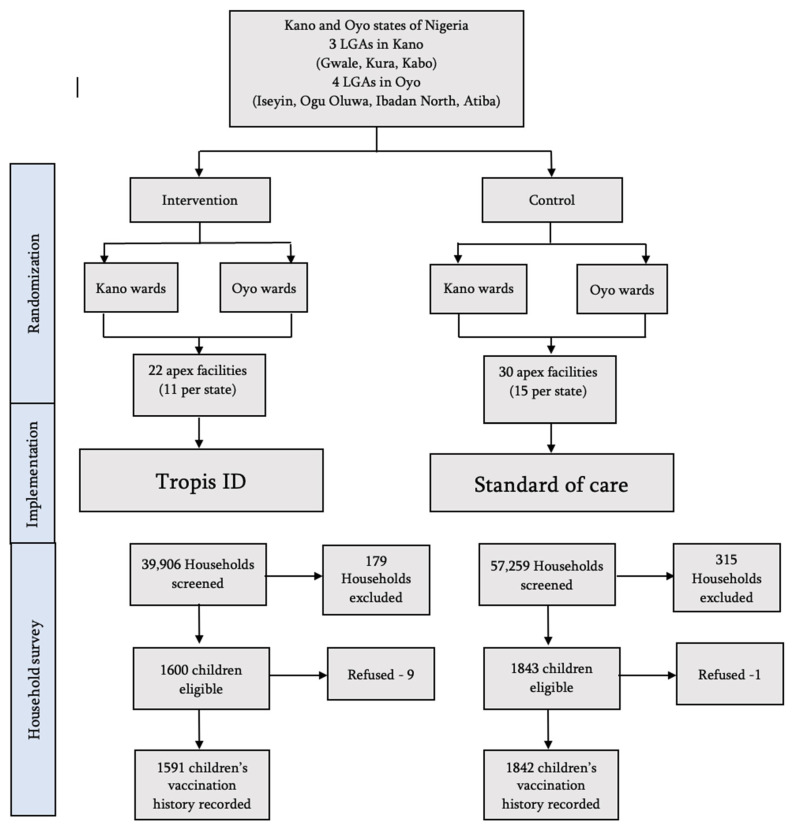
Trial flowchart.

**Figure 3 vaccines-13-00533-f003:**
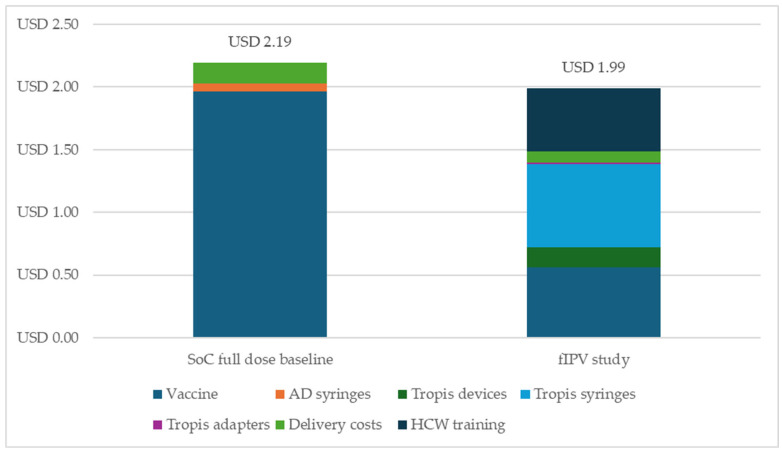
Cost per dose administered for fIPV versus standard of care under the study settings, according to 2023 USD. Abbreviations: AD, autodisable; fIPV, fractional inactivated poliovirus vaccine; HCW, healthcare worker; SoC, standard of care (full-dose IPV).

**Figure 4 vaccines-13-00533-f004:**
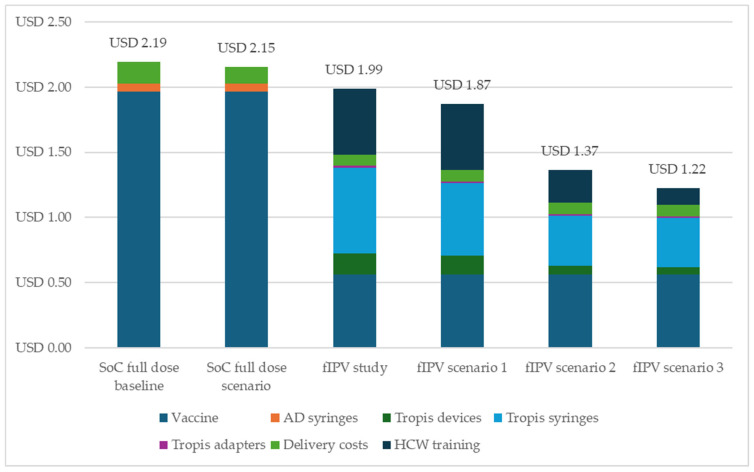
Cost per dose administered for fIPV versus standard of care as a result of scenario analysis, according to 2023 USD. Abbreviations: AD, autodisable; fIPV, fractional inactivated poliovirus vaccine; HCW, healthcare worker; SoC, standard of care (full-dose IPV).

**Table 1 vaccines-13-00533-t001:** Key unit prices and assumptions for the costing analysis.

Input	Full-Dose IPV (SoC)	fIPV Using Tropis	Data Source
IPV price per dose	USD 1.90	USD 0.38	UNICEF price for 10-dose IPV (2023 price)
Autodisable syringe price	USD 0.06	NA	
Tropis device price (reusable)	NA	USD 415, USD 374, USD 291	PharmaJet
Tropis syringe price (per dose)	NA	USD 0.59, USD 0.50, USD 0.34	PharmaJet
Tropis adapter price (one per vial)	NA	USD 0.45, USD 0.41, USD 0.38	PharmaJet
Shipping and clearance rates for vaccines	3.51%	3.51%	UNICEF
Shipping and clearance rates for autodisable syringes	3.51%	NA	UNICEF
Shipping and clearance rates for Tropis devices	NA	6%	PharmaJet
Shipping and clearance rates for Tropis syringes	NA	9%	PharmaJet
Shipping and clearance rates for Tropis adapters	NA	18%	PharmaJet
Wastage rate	15%	20%, 30%, 40%	Assumption: wastage rates for fIPV are incrementally more than those for the SoC

**Table 2 vaccines-13-00533-t002:** Receipt of two doses of inactivated polio vaccine among all respondents across the study arms (intention to treat).

	Control					Intervention					Overall
	n	%	95% CI		Number of Children	n	%	95% CI		Number of Children	
Total	609	63.8%	58.9%	68.8%	1842	982	61.7%	56.4%	67.0%	1591	3433
Kano	326	68.0%	61.5%	74.4%	1018	559	63.2%	56.9%	69.4%	885	1903
Oyo	283	58.7%	52.4%	65.0%	824	423	59.9%	50.9%	68.9%	706	1530

**Table 3 vaccines-13-00533-t003:** Receipt of two doses of inactivated polio vaccine among caregivers present during vaccination across the study arms (per protocol).

	Control					Intervention					Total
	n	%	95% CI		Number of Children	n	%	95% CI		Number of Children	
Total	1009	72.5%	67.6%	77.3%	1392	516	82.3%	76.4%	88.2%	627	2019
Kano	600	76.7%	71.5%	82.0%	782	239	89.2%	82.7%	95.7%	268	1050
Oyo	409	67.0%	60.1%	74.0%	610	277	77.2%	69.9%	84.4%	359	969

**Table 4 vaccines-13-00533-t004:** Regression model-based treatment estimates for absolute difference in the coverage for the two doses of inactivated polio vaccine.

	Intention to Treat (N = 3433)		Per Protocol (N = 2019)	
	Adjusted Coverage	Difference	Adjusted Coverage	Difference
Control	65.1% (61.5%,68.8%)	−5% (−11.7%, 1.7%)	71.9% (69.2%,74.7%)	11.2% (6.4%, 16.1%)
Intervention	60.1% (55.3%,65.0%)		83.2% (79.3%,87.1%)	

**Table 5 vaccines-13-00533-t005:** Procurement costs for IPV vaccines and supplies over a 5-year period in 2023 USD.

	Total over a Five-Year Period
**Full-dose IPV**	
Vaccines	USD 125,965,983
Autodisable syringes	USD 3,580,086
** Total cost of full dose**	**USD 129,546,069**
**Fractional dose IPV**	
Vaccines	USD 35,990,281
Tropis devices	USD 11,590,163
Tropis adapters	USD 719,806
Tropis syringes	USD 32,201,830
** Total cost of fractional dose**	**USD 80,502,079**
**Difference (savings from fractional dose)**	**USD 49,043,990**

## Data Availability

The data presented in this study are available on request from the corresponding author.

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
