# Peer review of "Evaluating the Impact of Needle-Free Delivery of Inactivated Polio Vaccine on Nigeria’s Routine Immunization Program: An Implementation Hybrid Trial"

_vaccines, 2025, doi:10.3390/vaccines13050533_

Round 1
Reviewer 1 Report
Comments and Suggestions for Authors
This is an excellent manuscript that introduces the Tropis needle-free device that is considered feasible and acceptable for use in Nigeria’s RI program for delivery of fractional doses of IPV alongside other co-administered vaccines.
The manuscript holds a new innovation and is very well written and organized.
However, I have some minor comments before accepting the paper for publication:
1- Why is the device not for sale ?
2- Please elaborate what are the skin layers that the device bypass when applied.
3- Also, please elaborate more on the target cells whether the dendritic cells or other cells that are found sub-cutaneous and usually contribute to the success of the vaccination through this route.
Author Response
Thank you for your thoughtful questions and comments. We have tried to clarify these below.
This is an excellent manuscript that introduces the Tropis needle-free device that is considered feasible and acceptable for use in Nigeria’s RI program for delivery of fractional doses of IPV alongside other co-administered vaccines.
The manuscript holds a new innovation and is very well written and organized.
However, I have some minor comments before accepting the paper for publication:
Comment 1 - Why is the device not for sale ?
Response 1 - The device is currently commercialized and “for sale”. Tropis has been sold to GPEI programs including WHO and UNICEF and US CDC in Somalia, Nigeria, Pakistan, and Afghanistan as well as Covid programs in India. The device is WHO prequalified
Comment2- Please elaborate what are the skin layers that the device bypass when applied.
Response 2 - The intradermal layer of the skin is the area between the epidermis (outer layer) and the dermis (inner layer). With Tropis, the thin fluid stream enters the skin through the epidermis just above the dermal layer creating an envelope or bubble, known as a bleb..
Comment 3- Also, please elaborate more on the target cells whether the dendritic cells or other cells that are found sub-cutaneous and usually contribute to the success of the vaccination through this route.
Response 3 - The intradermal layer:
- Is rich in diversity of APCs (Kupper et al, Nat Rev Immunol 2004)
- Induces long-lived Ab response (Levin et al., J Invest Dermatol, 2017)
- Promotes CD8+ T cell responses (Best et al., Vaccine 2009; Lind et al., Scand J Infect Dis, 2012; Duffy et al., Immunity, 2012 )
Reviewer 2 Report
Comments and Suggestions for Authors
In this paper, the authors evaluated the impact of needle-free delivery of inactivated polio vaccine on Nigeria’s routine immunization program by implementing a hybrid trial. This article is important because for many years Nigeria, especially the northern part, was faced with a polio pandemic, which escalated for several years until recently, when the country was declared polio-free. Below are my comments about this work
- The authors presented a cost analysis. What about cost-effectiveness analysis?
- The authors should clarify why they chose the states they considered in Nigeria.
- The authors should discuss the implications of their work since Nigeria has been declared polio-free.
Author Response
Thank you for the valuable comments and feedback on our study. Please find attached the responses to your comments. We have made changes with tracking in the manuscript.
In this paper, the authors evaluated the impact of needle-free delivery of inactivated polio vaccine on Nigeria’s routine immunization program by implementing a hybrid trial. This article is important because for many years Nigeria, especially the northern part, was faced with a polio pandemic, which escalated for several years until recently, when the country was declared polio-free. Below are my comments about this work
Comment 1 - The authors presented a cost analysis. What about cost-effectiveness analysis?
Response 1 - Thank you for this question. We did not conduct a cost-effectiveness analysis (CEA) because a CEA is only needed when an intervention is both more expensive and more effectiveness than the comparator and so the CEA would determine whether the incremental cost associated with the intervention is worth the incremental health impact. In our study, we found that fIPV with Tropis was actually cost saving and also more effective as it improved coverage, compared to SoC. Therefore, a CEA was not needed as the fIPV with Tropis dominated SoC. We have added this information into the discussion section for clarity on why a CEA was not needed.
Comment 2 - The authors should clarify why they chose the states they considered in Nigeria.
Response 2 - Two states were chosen one each in the north and south to balance the local stakeholder expectations in Nigeria. Both states were selected to receive the intervention based on the following pre-determined criteria: vaccination coverage of the third dose of pentavalent vaccine 40% to 60% (as a measure of the strength of the vaccination program), geographically acces-sible, conflict-free operating environment, and absence of other fIPV programs during the intervention period.
This information is added to the study setting section (methods) in the manuscript.
Comment 3 - The authors should discuss the implications of their work since Nigeria has been declared polio-free.
Response 3 - The implication is all the more important in that WHO goals for IPV1& 2 coverage are not being met in Nigeria. 89 cVDPV2 cases have been detected in the last 12 mo which is more than 2 times the next 2 countries combined. Many zero dose children in Nigeria leave country prone to reintroduction of wild type polio and to transmission and paralysis of VDPV.
Recent downshifts in funding for global health programs (including the Global Polio Eradication Initiative), have left programs seeking innovations with reduced costs but without losing program fidelity. Tropis has been shown in models prior to this study to decrease total immunization costs in campaigns and RI. If adopted, the cost savings could be reallocated to other high priority GH programs.
We have added this to the conclusions section in the manuscript.
Reviewer 3 Report
Comments and Suggestions for Authors
After reading carefully thre manuscript, I thought that this is a very meaningful job and hard work that benefits mankind. It is a nice paper about evaluation of the aTropis® ID device (PharmaJet®), a needle-free injection system, I do not find any questions based my review from it in language and logic and so on , and this revised vision shall be accepted. but I have some thoughts:
1, Control group shall be made between the aTropis® ID device (PharmaJet®), a needle-free injection system with conventional syringes with needles for Immune efficiency and cost
2,it is possible for conventional syringes with needles to carry some pathogens to transmit among different pepole
3, 5a, 6a? they shall be deleted
4, minor harm shall be mentioned using this system (Or precautions)
5,About the layout, The table shall not span across pages
6, I do not find the Summary content about 22 intervention facilities using Tropis for fIPV and 30 control facilities
7,in figure , one Vertical line shall be deleted
Author Response
Comments and Suggestions for Authors
After reading carefully thre manuscript, I thought that this is a very meaningful job and hard work that benefits mankind. It is a nice paper about evaluation of the aTropis® ID device (PharmaJet®), a needle-free injection system, I do not find any questions based my review from it in language and logic and so on , and this revised vision shall be accepted. but I have some thoughts:
Thank you for you kind suggestions. We hope this intervention will benefit mankind by providing a needle free experience for immunization.
1, Control group shall be made between the aTropis® ID device (PharmaJet®), a needle-free injection system with conventional syringes with needles for Immune efficiency and cost
The control group is the standard of care in Nigeria which is the use of conventional syringes to deliver vaccines intramuscularly. Hence the comparison has been made between the Tropis device and the conventional syringes with needles.
2,it is possible for conventional syringes with needles to carry some pathogens to transmit among different pepole .
We agree that conventional syringes with needles can transmit pathogens if safe inject practices are not adopted. We hope the Tropis device helps overcome some of these issues through their needle free approach.
3, 5a, 6a? they shall be deleted
The notations are to mark the location in the manuscript of the items from the SPIRIT checklist which is a checklist to follow for manuscripts describing randomized trials. We are complying with publication guidelines.
4, minor harm shall be mentioned using this system (Or precautions) _
During our study, we did not observe any side effects or harm with the TROPIS device. So we are unable to comment on the occurrence of any side effects.
We speculate that any side effects will be minor and similar in occurrence or less frequent than the use of conventional syringes with needles
5,About the layout, The table shall not span across pages
Thank you for pointing this out. We have added the tables within the manuscript template provided by the journal. During the proof-reading process, we will make adjustments to the table as requested by the journal.
6, I do not find the Summary content about 22 intervention facilities using Tropis for fIPV and 30 control facilities
The summary about the 22 intervention facilities and 30 control facilities are provided in the table in the appendix at the end of the manuscript.
7,in figure , one Vertical line shall be deleted
Thank you for highlighting. We will delete the line in the figure.